



# XLUM: an open data format for exchange and long-term data preservation of luminescence data

Sebastian Kreutzer[1,2,3], Steve Grehl[4], Michael Höhne[5], Oliver Simmank[5], Kay Dornich[5], Grzegorz Adamiec[6], Christoph Burow[7], Helen Roberts[1], and Geoff Duller[1]

[1]Geography & Earth Sciences, Aberystwyth University, Wales, United Kingdom
[2]Archéosciences Bordeaux, UMR 6034, CNRS - Université Bordeaux Montaigne, Pessac, France
[3]*present address: Institute of Geography, Heidelberg University, Heidelberg, Germany*
[4]HUK-Coburg, Coburg, Germany
[5]Freiberg Instruments GmbH, Freiberg, Germany
[6]Institute of Physics, Division of Geochronology and Environmental Isotopes, Silesian University of Technology, Gliwice, Poland
[7]piazza blu[2] GmbH, Cologne, Germany

**Correspondence:** Sebastian Kreutzer (sebastian.kreutzer@uni-heidelberg.de)

**Abstract.** Open data has become *the* modern science meme, and major funding bodies and publishers support open data. On a daily basis, however, the open data mandate frequently encounters technical obstacles, such as a lack of a suitable data format for data sharing and long-term data preservation. Such issue is often community-specific and best addressed through community-tailored solutions. In Quaternary sciences, luminescence dating is widely used for constraining the timing of event-based processes (e.g., sediment transport). Every luminescence-dating study produces a vast body of primary data that usually remains inaccessible and incompatible with future studies or adjacent scientific disciplines. To facilitate data exchange, long-term data preservation, in short, open data, in luminescence dating studies, we propose a new *XML*-based structured data format called *XLUM*. The format applies a hierarchical data storage concept consisting of a root node (node 0), a sample (node 1), a sequence (node 2), a record (node 3) and a curve (node 4). The curve level holds information on the technical component (e.g., photomultiplier, thermocouple). A finite number of curves represent a record (e.g., an optically stimulated luminescence curve). Records are part of a sequence measured for a particular sample. This design concept allows the user to retain information on a technical component level from the measurement process. The additional storage of related metadata fosters future data mining projects on large datasets. The *XML*-based format is less memory efficient than binary formats, however, in focus is data exchange, preservation and hence *XLUM* long-term format stability by design. *XLUM* is inherently stable to future updates and backwards compatible. We support *XLUM* through a new R package 'xlum', facilitating the conversion of different formats into the new *XLUM* format. *XLUM* is licensed under the MIT licence and hence available for free to be used in open and closed-source commercial and non-commercial software and research projects.





## 1 Introduction

Wilkinson et al. (2016) proposed four key principles for scientific data management towards open science: Findability, Acces-
sibility, Interoperability, and Reusability—the FAIR guidelines. Since then, major funding bodies (e.g., Thorley and Callaghan,
2019; Agence Nationale de la Rechereche (ANR), 2019; European Commission, 2021; Deutsche Forschungs Gemeinschaft
(DFG), 2022) and publishers (e.g., Copernicus Press Release, 2018; Wiley Author Service, 2022) have adopted these princi-
ples as part of their data management policies, and they have become an integral part of the European Code of Conduct for
Research Integrity (ALLEA, 2017). If interweaved with umbrella terms such as 'open data' or 'open science', the added value
of transparency and reproducibility of modern science comes across as almost self-evident. Unfortunately, the implementation
is often seems to fall behind set goals. For instance, Perkel (2020) vividly covered the challenge of 35 participants trying to
run decade old-computer code and concluded that maintaining reproducibility of software-based models and analysis pipelines
over decades is a demanding, sometimes impossible, task. Likewise, we can infer that data formats tied to a small number
of (outdated) programmes runs the risk that data becomes inaccessible. Another aspect on the data side considered by Noy
and Noy (2020) who complained that common open-data surrogate statements in articles such as 'data being available upon
request' may equate to no data access. Indeed, a pivotal aspect of the FAIR guidelines is their emphasis on principles fostering
**automated** data processing or enabling such processing in the first place. The requirement to actively contact the study authors
to request access to the data, e.g., e-mail requests, therefore inherently undermines the principles of open data(Noy and Noy,
2020). On the other hand, authors perhaps refrained from direct sharing because of unclear reporting guidelines or the effort
required to document data of presumed low demand.

Adhering to the FAIR guidelines with actual benefits for all parties (e.g., data donators, data users, funding bodies) involves
tackling low-level technical issues, such as defining an exchange data format enabling study authors to share their raw data
in a manner which is structured, standardised, and ideally effortless, and in a format that will remain accessible long into the
future. Here we adopt the idea that those issues are usually community-specific and best addressed through discipline-tailored
solutions, for instance, for data generated in luminescence-based chronology studies.

Luminescence dating is a dosimetric dating method of key importance in Quaternary sciences and archaeology (e.g., Rhodes,
2011; Roberts et al., 2015; Bateman, 2019), covering around the last 300 000 years. In a nutshell, the datable event is the last
sunlight or heat exposure of natural minerals such as quartz or feldspar. The dating process determines two parameters: (1)
the absorbed dose (in Gy) accumulated in the minerals since the last heat or light exposure, and (2) the environmental dose
rate (in $Gy\,ka^{-1}$). The ratio of dose (Gy) divided by dose rate ($Gy\,ka^{-1}$) gives the age (ka). Methods frequently applied in
luminescence dating studies are distinguished by their stimulation mode, e.g., thermally stimulated luminescence (TL; cf.
Aitken, 1985), optically stimulated luminescence (OSL; Huntley et al., 1985) or infrared stimulated luminescence (IRSL; Hütt
et al., 1988). Luminescence methods are also used by adjacent scientific disciplines, e.g., accident dosimetry and material
characterisation (e.g., Yukihara and McKeever, 2011; Yukihara et al., 2014).
Luminescence (dating) does not measure the absorbed dose directly but infer an equivalent dose ($D_e$) from the minerals'
natural light output (luminescence) compared to a laboratory dose of known size. Luminescence dating studies and research





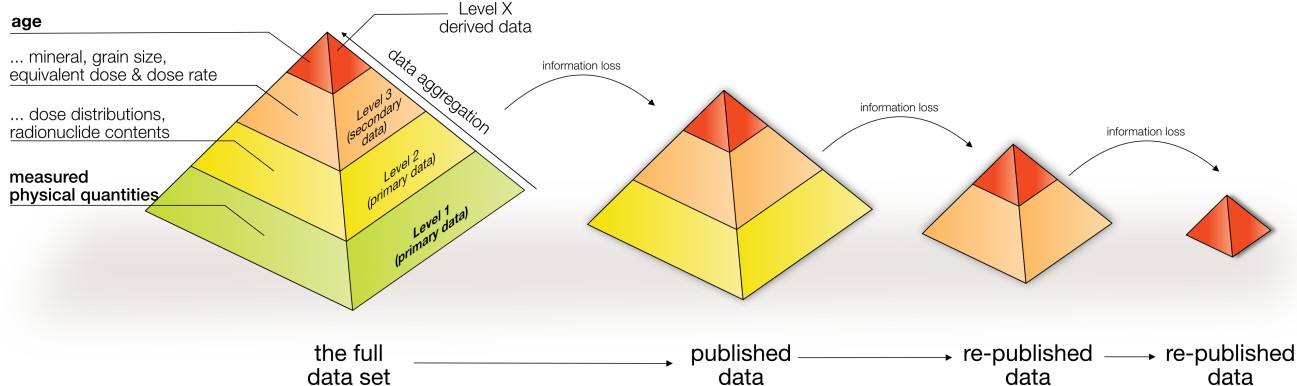

**Figure 1.** A luminescence age is the result of data aggregation. In order to reproduce all steps, access to primary data (the base level) is indispensable. However, such primary data are seldom published or otherwise accessible. Re-publishing usually leads to information loss. The number of information/process levels in the graph is arbitrary.

building on such work routinely tabulate only a fraction of the recorded data in the form of aggregated parameters. One could think of a pyramidal information hierarchy Fig.1 with the age on the top. The base is made out of minimally processed luminescence data, i.e. measured luminescence (for the purpose of this manuscript we "neglect" the dose rate information).

Original dating studies ideally report the full information pyramid. However, the further an age is carried forward through subsequence studies or collected in data repositories, the higher the level of data aggregation. Good examples for aggregated luminescence data are repositories such as Lancaster et al. (2015) or Codilean et al. (2018). Such archives are excellent places to find locations of dating studies, but it is not easy to spatially link different ages without accessing the original studies with primary data.

Original, minimally processed luminescence data (see Fig.1), i.e. measured luminescence, is hardly ever published along with a study. However, sharing of unprocessed luminescence data, accessible to others after the completion of a dating study, is desirable for several reasons:

1. Luminescence ages are end-members of long measurement series involving various protocols, tests, and analysis steps with potentially different hardware and software tools. Once aggregated, it is challenging for others to re-validate pub-

lished luminescence dates beyond plausibility checks. Shared raw data will potentially lead to better reproducibility and data quality.

2. Access to luminescence data on a single curve level supports the application of advanced analysis tools employing hierarchical Bayesian models such as the R package `BayLum` (Philippe et al., 2019) or the model 'baSAR' (Combès et al., 2015; Mercier et al., 2016). Other work has shown examples of how to study sediment pathways by tracing

the bleaching histories of sediment grains (Chamberlain and Wallinga, 2019). If such data are never shared, their full potential remains untapped.



3. Recently, Balco (2020) advocated for a transparent and open middle-layer concept, disconnecting measured quantities from processed ages to account for changed, perhaps improved calculation procedures. His proposal was specified to cosmogenic-nuclide exposure dating, but the general idea appears valid for other dating techniques, such as luminescence dating. For instance, it would enable others to test the impact of alternative applied statistical parameters on the calculated $D_e$ in the future.

4. The approach of Balco (2020) renders ages moving targets, i.e. they may change with time due to different calculation procedures. Balco's approach emphasis the data treasure character of measured physical quantities (with "data are described with rich metadata" FORCE11 2014), which need to be preserved and shared instead of processed numbers. This approach holds for luminescence dating studies, which create, somewhat as a by-product, a vast amount of luminescence data of minerals from different origins. Such data are of potential interest, for instance, to geoscientists working on provenance analysis (e.g., Sawakuchi et al., 2018; Tsukamoto et al., 2011), to physicists focusing on luminescence models, or to data scientists trying to develop new approaches to enable exploratory luminescence data analysis to constrain physical parameters of OSL curves (e.g., Burow et al., 2016), or seeking training datasets to test machine learning approaches (e.g., Kröninger et al., 2019).

5. Broadly shared and accessible through a standard format, luminescence curve data will help establish a comprehensive repository for luminescence data, enabling studies and meta-studies not covered by the above mentioned examples.

Data sharing requests can only be reasonably accommodated if luminescence data can be easily exchanged, sufficiently archived, and analysed independently of proprietary software or file formats. We argue that one particular reason hampering the exchange and reuse of luminescence data is the absence of a suitable data format supporting long-term data preservation and fostering data exchange. To our best knowledge, long-term data preservation is an unresolved issue in the luminescence (dating) community. After being analysed and published, one can expect original primary data to be archived in compliance with scientific standards, but they may become inaccessible or incompatible with new data over time when the re-analysis is wanted. Such data are often lost to the public and need to be measured again. Hence, the first step of chronological data sharing and archiving is a data format that qualifies to serve that purpose.

In this contribution, we first briefly list existing data formats commonly used to store luminescence data. We then outline identified general technical requirements for a data format for the long-term preservation of luminescence data. Hereafter, we highlight features of a new *XML*-based file format, *XLUM*, developed for long-term preservation and exchange of luminescence data. The remainder provides examples and illustrates a reference implementation in R and Python, showing how existing data can be converted effortlessly into the new *XLUM* format. The discussion addresses potential shortcomings and challenges and canvasses future directions. We consider our contribution as an initial definition, and the format blueprint is open to discussion within the luminescence community.

The *XLUM* file format is published and fully detailed on a *GitHub^TM* repository and archived on *Zenodo* (European Organization For Nuclear Research and OpenAIRE, 2013) (Kreutzer et al., 2022b). The chosen open-source MIT licence (https://opensource.org/licenses/MIT, last accessed: 2022-05-2) allows reuse in open-access and closed-source software projects.





At last some format conventions, hereafter we will use `monospace` letters for format/code snippets, and file format arguments. *XML*-elements (nodes), if not accompanied by a closing tag are contracted into one short tag, for instance, `<node/>` instead of `<node> ... </node>`.

## 2 Existing data formats in the luminescence-dating community

Equipment manufacturers have introduced most output data formats available in the luminescence-dating community. For instance, *Daybreak* (Bortolot, 2000), *lexsyg* (Freiberg Instruments, Richter et al. 2013, 2015), *Risø TL/OSL reader* (e.g., Bøtter-Jensen, 1988, 1997; DTU Nutech - Center for Nuclear Technologies, 2016), *SUERC portable OSL* (Sanderson and Murphy, 2010). Alternative formats were developed as part of research studies (e.g., Mittelstraß and Kreutzer, 2021). In other cases of equipment development, data output formats were not mentioned explicitly (Markey et al., 1997) or the hardware relied on

export options of commercial laboratory software solutions (Guérin and Lefèvre, 2014; Mundupuzhakal et al., 2014). Some file formats are proprietary, most are not documented in full. Additionally, data stored in comma-separated value files (file extension *\*.csv*) or raster image-file formats (*\*.tif*, *\*.spe*) appear to be common, however, they lack the metadata required for luminescence data analysis.

**Table 1.** List of file formats dedicated to store luminescence data in alphabetic order (non-exhaustive).

| File extension | Type | Relation |
|---|---|---|
| `*.bin/*.binx` | binary | Risø TL/OSL readers |
| `*.dat` | binary | Daybreak |
| `*.psl` | ASCII | SUERC portable |
| `*.rf` | ASCII | Mittelstraß and Kreutzer (2021) |
| `*.txt` | ASCII | Daybreak |
| `*.xsyg` | ASCII | lexsyg |

Because (proprietary) file formats serve a particular purpose and equipment setup, the information stored in such files varies greatly

across the different neatly tailored formats. Furthermore, they are incompatible in the sense that information transfer from one file format to another causes information loss (lossy coercion). Very popular is the BIN/BINX-format, which is well documented with good support from other manufacturers (e.g., Freiberg Instruments) and through community-maintained software solutions such as the *Analyst* (Duller, 2015), *LDAC* (Liang and Forman, 2019) or the R (R Core Team, 2022) packages *'Luminescence'* (Kreutzer et al., 2012) or *'numOSL'* (Peng et al., 2013).

The BIN/BINX-format was introduced decades ago but it is not the most suitable candidate for long-term data preservation and exchange because: (1) Different file format versions are incompatible because of non-identical file header lengths and byte order; (2) Storage of additional, so far unspecified, metadata requires a format change triggering a new format version. (3) For historical and memory-efficient reasons, instead of xy-data, only y-data (here counts per channel) are stored, and the



temperature is deduced linearly from maxima and the minima values (see Fig. 2). A thermoluminescence (TL) curve repre-

sents luminescence against stimulation temperature indeed. However, a detection system usually consists of two independent technical components. One records the luminescence signal, e.g., a photomultiplier tube (PMT), and the other monitors the temperature, e.g., a thermocouple. Both quantities are recorded as a function of time, not temperature. (4) Data repositories should be findable (e.g., unique identifiers, proper metadata, cf. Wilkinson et al. 2016, Box 2, p. 4 ), and accessible by standard parser libraries (e.g., *libxml*), and the requirement of format-tailored software solutions should be avoided.

The other data formats listed in Table 1 suffer from similar or related problems because they were designed to accommodate data for a sole purpose or limited application range. In contrast, what is arguably preferable is a format that is as accessible and findable as possible and independent of a specific type of equipment; a requirement that laid the foundation for the development of *XLUM*.

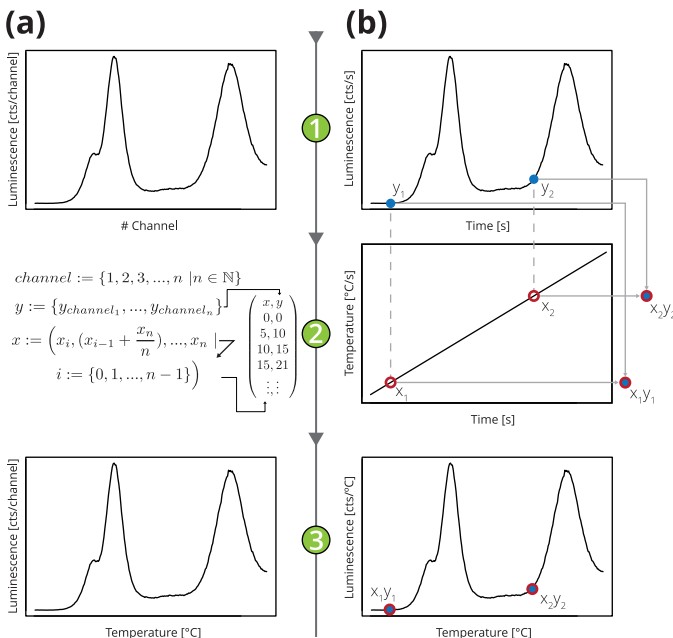

**Figure 2.** Simplified illustration of two approaches to store a typical TL curve. (A) in the "conventional way" count data of the PMT are recorded channel-wise (1) and the temperature values are re-calculated (2) based on the minimum and maximum values to obtain the (3) final TL curves while data is imported in a programme. (B) in the approach suggested here, the luminescence signal and the temperature, are recorded by two independent technical components, e.g., a PMT (1) and a temperature sensor (2) monitoring the heating process. While importing, the resulting TL curve (3) becomes a match of both recorded signals on the time domain. Such import routine is, e.g., available in the R package 'Luminescence' (Kreutzer et al., 2012).



## 3 General data format requirements

A few design prerequisites guided the development of the *XLUM* format, and we list the most important below.

- The format should preserve physical quantities (measured, modelled), while, and their description remains equally readable to humans and machines.

- Data should be stored structured on a technical component/sensor level (e.g., photomultiplier, thermo-couple) without limiting the data or forcing data reduction.

- The format shall enable self-contained storage of data from technical components.

- The format is self-explanatory, i.e., it can be generally understood without format documentation.

- Backwards compatibility shall be maintained for future versions (newer format versions may carry additional attributes but remain readable to existing tools).

- The format provides a neutral format open and "non-proprietary" with specifications defined by the scientific community
not by equipment manufacturers.

- The format application is permitted in closed and open software tools through suitable license conditions.

- Standard software solutions to process measurement data, e.g., *MS Excel[TM]*, R, Python, *LibreOffice*, *Matlab[TM]*, *GNU Octave* are to be supported.

- Data preservation and exchange should be facilitated independent of the operating system running on users' personal
computers.

- The FAIR guidelines are supported by design.

- The format shall facilitate the creation of large repositories for long-term preservation and exchange of luminescence measurement and metadata.

We identified an XML (Extensible Markup Language)-based (W3C XML Core Working Group, 2008) format as the most
suitable structure serving the outlined requirements.

The idea of introducing an XML-based format for storing luminescence data is not new. Bortolot and Bluszcz (2003) have sketched a few general requirements for such a format nearly 20 years ago, although this approach has not been widely adopted. An *XML*-based format is rather memory inefficient, particularly if compared to binary formats, leading to relatively large files (tens of megabytes and more instead of megabytes). However, we believe that this aspect is of limited relevance because: (1)
mass data storage is inexpensive, particularly if costs are compared to 2003, the year of the article by Bortolot and Bluszcz (2003); (2) The overall amount of data produced in luminescence dating is negligible compared to other disciplines working with *XML*-based formats (e.g., Martens et al., 2011; Röst et al., 2015); (3) Modern storage systems of data repositories usually

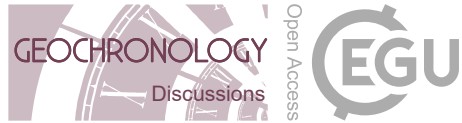

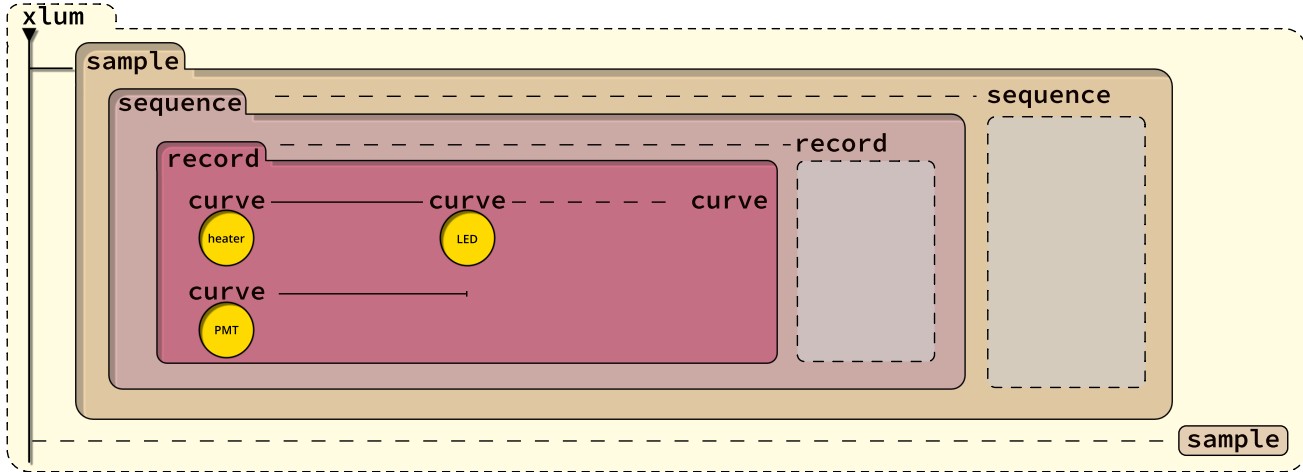

**Figure 3.** The figure is a graphical representation of the data storage concept with the different node levels of the *XLUM* format. Data are stored sequentially over time. Dashed lines are used to indicate the possibility of multiple instances. For example, one *XLUM* file can contain many `<xlum/> nodes` and one `<xlum/>` node many `<sample/>` nodes etc.

employ highly efficient low-level data compression methods (e.g., lossless data compression) independent of any file format, reducing the data footprint regardless of the exchange format.

## 4 Format description

In the following, we outline the conceptual structure of the *XLUM* format. To minimise verbosity, we only focus on key design concepts. For full details, we refer to our reference document on *GitHub^{TM}* (https://xlum.r-luminescence.org, last accessed: 2022-06-30). The *GitHub^{TM}* repository also contains a formal format description following the XML Schema Definition (XSD) for automated validation. *XLUM* defines a substructure, which can be part of a file or any other XML structure (W3C XML Core Working Group, 2008) that acts as a container or constitutes a file of its own, for instance, with the file extension *.xlum*. Although the *XLUM* format does not enforce a specific file extension.

The two key features of the *XLUM* format are (1) information nesting, with measurement data only stored in the lowest node, (2) support of data sharing by design.

### 4.1 Nesting of information on five node levels

The format consists of five levels (nodes) (Listings 1, Fig. 3), indicated by so-called tags. The correct formal description requires an opening tag (`<...>`) and a closing tag (`</...>`) (see Listings 1). Each tag allows various attributes (`<tag attribute='' ...>`) for metadata of which we will detail a few below. The number of attributes is not limited, and additional user-defined attributes, not covered by the format definition, are explicitly allowed. However, the format definition





insists on mandatory attributes, a few accepting the non-empty string NA for not available/not applicable. The first upper four nodes structure the data and provide metadata to describe the dataset. The lowest node (`<curve/>`) contains the raw (or minimal processed) measurement data.

**Listing 1.** 'Basic hierarchical structure of the *XLUM* format following the XML scheme in version 1.0 with UTF-8 encoding. The three dots (`...`) indicate node attributes.'

```
 1: <?xml version='1.0' encoding='utf-8'>
 2: <xlum ...>
 3:   <sample ...>
 4:    <sequence ...>
 5:     <record ...>
 6:      <curve ...>
 7:        12 23 23 23 13 23
 8:      </curve>
 9:     </record>
10:    </sequence>
11:   </sample>
12: </xlum>
```

1. **`<xlum/>`** is the root node. It wraps all other data and is parent to all other child nodes. The number of child nodes of `<xlum/>` is unlimited. Everything within one `<xlum/>` is considered a collection of data for different samples for which, e.g., author names, digital object identifier (DOI), and a license can be assigned through attributes.

2. **`<sample/>`** is the first child node to `<xlum/>`. It is the parent structure for luminescence data collected for a single sample. Hence, everything wrapped between `<sample/>` refers to a specific sample. Amongst others, expected attributes are the `name` of the sample and the geographic coordinates (`latitude`, `longitude`).

3. **`<sequence/>`** is the first child node to `<sample/>`. It sets the structure for measurement data defined through (measurement) sequences, e.g., a single aliquot regenerative (SAR) dose (Murray and Wintle, 2000) measurement sequence or any measurement data arranged in a particular order. Typical attributes are `position`, `fileName`, or `readerName`.

4. **`<record/>`** holds all records of a sequence of a particular sample. A record is not necessarily limited to a single measured xy-curve but a collection of xyz-data created by technical components, which together define a record, e.g., a TL measurement. A typical argument is `recordType`.

5. **`<curve/>`** is the lowest node and the only node containing measurement data. All data stored in this node refer to a single technical component, e.g., a photomultiplier. One or many curves define one record. Numerical values in this node are separated by whitespace and span an array with three dimensions (see Eq. 1). Alternatively, this node allows data encoded as base 64 strings.



The crucial concept of the format is that data are stored only in `<curve/>` nodes, defined by technical components (actual or virtual) **measuring or simulating physical quantities over time**. Data in `<curve/>` are numerical (measurement/simulation) values of a physical quantity $v_1, ..., v_n; n \in \mathbb{R}$ (discrete/continuous) spanning an array of the form

$$A_{[x \times y]_1}, ..., A_{[x \times y]_t} \mid x, y, t \in \mathbb{Z}; \tag{1}$$

with $n = max(x) \times max(y) \times max(t)$, where $t$ is the extension of the array with respect to time (i.e. channels per time instant) and $x$ and $y$ define the lateral geometry of the detector. Data are stored column-wise starting from $A_{(1,1)_1}, A_{(2,1)_1}, ..., A_{(x,y)_1}$ before continuing in the time dimension. For instance, for measurement over 100 channels with a photomultiplier tube $x = y = 1$ and $t_1, ..., t_{100}$. In contrast, for a measurement with a camera with a lateral resolution of $512 \times 512$ pixels, $x, y \in 1, ..., 512$, where $t_1, ..., t_{100}$ remains the same. The dimensional information is stored in the node attributes `xValues`, `yValues`, and `tValues`. All quantities, except for $t$ are dimensionless (i.e., they have no default unit). However, the attributes `xUnit`, `yUnit`, and `tUnit` allow setting SI units. For more attributes and their meaning, we may refer to the detailed format description (https://github.com/R-Lum/xlum_specification, last accessed: 2022-07-04).

## 5   Data representation: example

To illustrate data storage in the *XLUM* format, we will pick one TL, and one green stimulated luminescence (GSL) record belonging to a test sequence measured for one sample. For simplicity, we limit the number of values for each curve to ten and substitute `NA` with "`...`" (three dots). We provide the complete file that can be imported correctly as a supplement.

**Listing 2.** Example luminescence-data representation in the *XLUM* format.

```
 1: <?xml version="1.0" encoding="utf-8"?>
 2: <xlum xmlns:xlum="http://xlum.r-luminescence.org" lang="en"
 3:    formatVersion="1.0" flavour="generic"
 4:    author="Marie_Sklodowska-Curie;_Max_Karl_Ernst_Ludwig_Planck"
 5:    license="CC_BY_4.0" ...>
 6:  <sample name="LUM-21321" mineral="quartz"
 7:    latitude="52.4091392" longitude="-4.0702446" altitude="50" doi="valid_DOI" ...>
 8:   <sequence position="1" name="Example"
 9:     fileName="Testsequence.seq" software="DeviceEditor_2.0" ...>
10:    <record recordType="TL" sequenceStepNumber="1" ...>
11:     <curve component="thermocouple" startDate="2021-02-14T22:57:12.0Z"
12:       curveType="measured" duration="10" offset="0" xValues="0"
13:       yValues="0" tValues="1_2_3_4_5_6_7_8_9_10" tLabel="time"
14:       vLabel="temperature" xUnit="" yUnit="" vUnit="K" tUnit="s" ...>
15:        293 303 313 323 333 343 353 363 373 383
```





```
16:        </curve>
17:        <curve component="PMT" startDate="2021-02-14T22:57:12.0Z"
18:          curveType="measured" duration="10" offset="0" xValues="0"
19:          yValues="0" tValues="1 2 3 4 5 6 7 8 9 10" tLabel="time"
20:          vLabel="luminescence" xUnit="" yUnit="" vUnit="cts" tUnit="s"
21:          detectionWindow="375" filter="Hoya U340; Delta BP 365/50EX" ...>
22:           100 210 320 450 560 700 800 900 850 650
23:        </curve>
24:      </record>
25:      <record recordType="GSL" comment="standard green OSL step"
26:        sequenceStepNumber="2" ...>
27:        <curve component="PMT" startDate="2021-02-14T22:57:20.0Z"
28:          curveType="measured" duration="10" offset="0" xValues="0" yValues="0"
29:          tValues="1 2 3 4 5 6 7 8 9 10" tLabel="time" vLabel="luminescence"
30:          xUnit="" yUnit="" vUnit="cts" tUnit="s" detectionWindow="375"
31:          filter="Hoya U340; Delta BP 365/50EX" ... >
32:           0.9 0.82 0.74 0.67 0.61 0.55 0.50 0.45 0.41 0.37
33:        </curve>
34:      </record>
35:    </sequence>
36:  </sample>
37: </xlum>
```

**Line 1** Mandatory entry, announcing the *XML* format. The format follows the Unicode® (The Unicode Consortium, 2022) UTF-8 and must not be changed. In a nutshell, it tells programs for file parsing the character encoding and ensures that characters are interpreted correctly.

**Lines 2–5** Start of the *XLUM* record, with mandatory entries, e.g., for the namespace (`xmlns::...`) and the used format version (here: `1.0`), and metadata related to the data itself, e.g., `author` and `license`. Those attributes apply to all child nodes and clarify the data sharing rights in simple and unequivocal terms. In the example, we have applied the Creative Commons (CC) Licence CC-BY[1]. This licence allows unrestricted data reuse, mixing, and sharing with the requirement to credit the data creators.

**Lines 6–7** The `<sample/>` node allows providing information about the sample, e.g., `mineral`, `latitude`. Those data are helpful for explorative data analysis with data from different geographical origins.

**Lines 8-9** The `<sequence/>` node, with general information that remains unchanged for a sequence, e.g., `position` for referring to a position in the equipment.

---

[1]https://creativecommons.org/about/cclicenses/, last accessed: 2022-11-26





**Line 10** The first record in the dataset, here of type `TL`.

**Lines 11–16** The complete entry for data as recorded by the thermocouple, i.e., the sensor monitoring the temperature of the period of 10 s (`tValues`, `tUnit`) recorded in K (`vUnit`). The data (temperature values) are the content of the `<curve/>` node in line 15. For the other two curves, the measured values are listed in line 22 (PMT TL curve), and line 32 (count values GSL).

**Lines 17-23** The 2$^{nd}$ `<curve/>` node with values measured with a PMT during heating (`component`). Here attributes for `detectionWindow` and `filter` are set. However, those attributes can be set to `NA`, if information is not available or not applicable.

**Lines 25–35** The 2$^{nd}$ record in the sequence with one file with GSL measured with a PMT. While the record may contain information about the stimulation power or settings of the heating during stimulation, that information was not recorded for our example.

**Lines 35–37** The closing tags for the nodes `<sequence/>`, `<sample/>` and `<xlum/>`.

The example shows that a considerable part of the information refers to data giving information about the data measured with a technical component (PMT or thermocouple), rendering the dataset self-contained and portable without requiring additional documentation.

## 6 Format support in R and Python

Beyond a community-tailored open data solution for storage and sharing, an open exchange format expands the availability of luminescence data beyond the target community. This enables experts from other disciplines to work with luminescence data. Data analytics is a field in the hemisphere of Artificial Intelligence, Machine Learning, and Data Science. Here, two programming languages have gained significant popularity R (R Core Team, 2022) and *Python$^{TM}$* (Python Software Foundation, 2022). To foster the distribution and usage of *XLUM*, we developed two prototype interfaces, one for R, and one for *Python$^{TM}$*.

### 6.1 Programming environment R

We realised support for the *XLUM* format for the statistical programming language R (R Core Team, 2022) through a new package called *'xlum'* (Kreutzer and Burow, 2022). The package has reached a stable development status with a test coverage of nearly 100 %, i.e., package functionalities are continuously tested in unit tests (cf. Kreutzer et al., 2017, for more details) on different operating systems. Within the R environment, data elements can be accessed through base R functionality and data types (e.g., `list`; see documentation of the package *'xlum'* and *'xml2'*).

In *'xlum'*, all file interactions (functions `read_xlum()`, `write_xlum()`, `validate_xlum()`) are realised through the R package *'xml2'* (Wickham et al., 2021) interfacing the *XML* parser library *libxml2* (https://gitlab.gnome.org/GNOME/libxml2, last accessed: 2022-07-05). The generation of *XLUM* files (function: `write_xlum()`) follows the workflow outlined





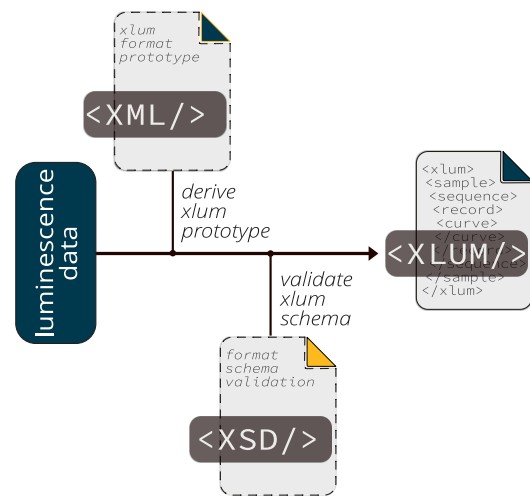

**Figure 4.** The workflow to generate *XLUM*-files as implemented in the R package *'xlum'*.

in Fig. 4. First, a format blueprint is derived from a prototype shipped with the package *'xlum'*. The prototype is then expanded

and filled with data. Before export, the file is validated (`validate_xlum()`) against an *XSD* schema to ensure that the produced *XLUM* file follows the correct format specification. Both, the prototype and the *XSD* are copies of files available as part of the *XLUM* file format definition.

Other than basic import and export functionality to and from R of data shared as *\*.xlum* files, the package *'xlum'* supports the conversion of various file formats (e.g., *\*.binx*, *\*.xsyg*) to the here proposed *XLUM* format using R functions commencing

with `convert_` (e.g., `convert_binx2xlum()`, see Fig. 5). Internally, the conversion uses the R package *'Luminescence'* (Kreutzer et al., 2012, 2022a) to create R specific object structures (S4-class objects) tailored to luminescence data called RLum-objects (cf. Kreutzer et al., 2017, their Fig. 2), and convert these structures using `convert_rlum2xlum()`. To date, this conversion is not always lossless, i.e., not all metadata are transferred to the *XLUM* format. The support will improve with the maturity of *XLUM*. For instance, the conversion of a *\*.binx* file requires the following R code lines:

**Listing 3.** 'Converting a *\*.binx* file into an *\*.xlum* file.'

```
1: library(xlum)
2:
3: convert_binx2xlum(
4:   file = 'test.binx'
5:   out_file = 'text.xlum')
```





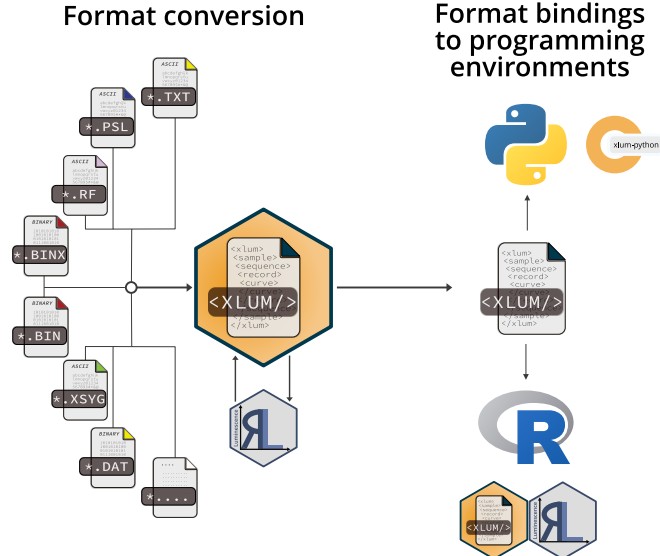

**Figure 5.** The R package 'xlum' supports the conversion of various commonly used luminescence (dating) data formats to `<xlum/>` using the R package 'Luminescence'. Bindings to the statistical programming language R and the general-purpose programming language Python are realised through language-specific software packages.

## 6.2 Python

Similar to R, Python is an interpreted language. It is beginner-friendly and viral outside of traditional software development and computer science. A major advantage is the large and active open source community maintaining a wide vari-
ety of packages (e.g., 'pandas', 'matplotlib', 'plotly') supporting data analysis workflows. For analysing luminescence data with Python, we provide a work-in-progress version of a package called 'xlum-python' on *GitHub*[TM] (https://github.com/ SteveGrehl/xlum-python, last accessed: 2022-07-01). The package allows loading *XLUM* files with Python and conversion into pandas DataFrame objects (two-dimensional tabular data). This format is a starting point for further analysis, such as conversion to CSV files, export to Microsoft *Excel*[TM] or graphical output. We show a minimalistic example of data im-
port using Python in Listings 4. We provide more information and examples on the corresponding *GitHub*[TM] repository (https://github.com/SteveGrehl/xlum-python, last accessed: 2022-07-01).

**Listing 4.** Minimal script for the usage of xlum-python

```
1: import xlum.importer
2: from xlum.data.classes import XlumMeta
3:
4: obj: XlumMeta = xlum.importer.from_xlum('path_to_/xlum_example.xlum')
```

Figure 6 shows a simple representation of the measured values from an example file.



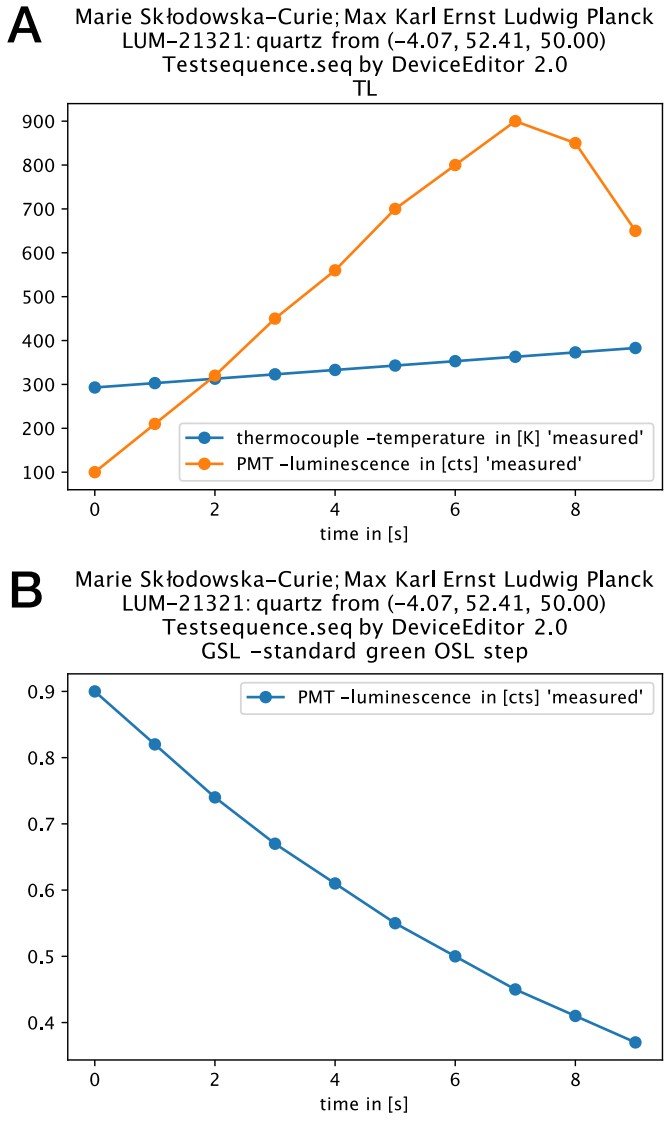

**Figure 6.** Graphical representation of an *XLUM* example file, imported with 'xlum-phyton' and visualised using the Python library 'matplotlib'.



## 7 Discussions

Our contribution aims to standardise luminescence data exchange and enable long-term data preservation by introducing a unified data format. Still, our initiative should not be interpreted as an attempt to abolish other existing formats. While *XLUM* is highly flexible, and broad support beyond the presented first implementation in R and Python is likely, all other formats have their justification. In particular, if considered as actual primary data, conversion to *XLUM* involves data coercion to some extent. However, format support through other software and equipment manufacturers is desirable in the long run, making

luminescence data more findable and accessible. *XLUM* promotes the implementation of the FAIR data sharing guidelines. Nonetheless, it does not enforce them, and our contribution should not be understood as a claim on how and if data should be shared. Instead, here we refer readers to the guidelines of their institutes or funding bodies.

For *XLUM*, we have chosen an *XML*-derived format structure. It is usually less memory efficient than any binary format, which we see as an acceptable weakness if it helps to facilitate human readability. During the specification process, we eval-

uated other similar structured data exchange formats, such as JavaScript Object Notation (*JSON*) (https://www.json.org/) or *YAML* (https://yaml.org). In particular, discussions about the advantages and disadvantages of *XML* vs *JSON* are considered in numerous IT websites, blogs, platforms such as Stack Overflow (https://stackoverflow.com), or in technology magazines. Without delving into their technical details, *JSON* has gained popularity over *XML* in recent years (e.g., Andy Patrizio, 2016). Nevertheless, given the widespread use (e.g., Copernicus Publications, 2014), and support of *XML* schemas for data repre-

sentation in various fields (see examples in Nolan and Lang, 2013), we opted for *XML* as a robust basis. *XML* provides a standardised grammar but remains flexible enough to be tailored to our purpose. If wanted and needed, luminescence data will remain easily transferable to other data formats once standardised and archived as *XLUM* files. Another possibility is an amendment of *XLUM*, for instance, to better facilitate image data, for which storage is already possible today, and the optional support of base 64 string encoding enables a more efficient representation of those data.

Throughout the manuscript, we implicitly carried forward the limitation that the *XLUM* data format concerns luminescence data only, albeit a luminescence age is obtained from a luminescence-derived equivalent dose divided by a dose rate. The latter term is as essential in luminescence dating as the equivalent dose, although our attention reflects somewhat the typical focus of luminescence-dating studies. Specifically, nearly every luminescence-dating laboratory has access to luminescence readers (recording luminescence data) with comparable technical capabilities. Those readers enable accurate and precise records

of dim light emissions down to a single-grain level. Protocols and methods differ. However, the primary data is luminescence (light) in all cases. In contrast, radio-nuclide concentrations values used to calculate dose rates are derived using different methods, e.g., high-resolution $\gamma$-ray spectrometry, in-situ $\gamma$-ray spectrometry, alpha/beta-counting, or inductively coupled plasma mass-spectrometry. Different methods make it challenging to develop a data format applicable to all methods. Still, a concept development, perhaps again with a focus on luminescence dating, might be a reasonable attempt.

The notion of open data is access and insight. However, shared data are not becoming automatically accessible, and not every data set may provide similar valuable insight but depend on the experimental design. One may argue that making data accessible instantaneously with each study published is advantageous for data users and disadvantageous for donors. For instance,



Mills et al. (2015) discuss potential adverse effects on the availability of primary data in the field of biology if investigators of long-term studies are obliged to share their data. Although *XLUM* is merely a data format, setting no sharing rules and our comparison with a different discipline quickly wears off. The fear of study authors that others may use their hard work to publish more quickly might be one of the reasons for the "upon reasonable request" data availability statements (Sec. 1). In the case of luminescence studies, long-term studies running over many years are scarce (e.g., Guérin and Visocekas, 2015, for an excellent example of such a study), and single datasets, even such as from a whole stratigraphic section as typical in palaeoenvironmental results studies, are of limited use to others. The true benefit of data sharing lies within many accessible and findable single datasets meaningfully linked through metadata, altogether forming large datasets. Because of its component focussed design with a minimum number of required metadata, *XLUM* does the groundwork for datasets aligned in luminescence-based chronologies across different sites in data mining projects concerned with luminescence model development and validation or in any explorative data analysis study.

Last, a significant obstacle to our initiative towards success is the question of broad community acceptance of the new format. Reasonable predictions are difficult to make. We tried to improve the chance of success of our initiative by implementing the first support in the programming language R and Python and by keeping all documents open-access. Furthermore, with the publication of the manuscript after peer-review, the *XLUM* format will be supported by *LexStudio2*, the software running *lexsyg* luminescence readers and could be supported by/adopted by other luminescence and dosimetry manufacturers. Further versions of this format will be developed transparently using the *GitHub*<sup>TM</sup> repository and are open to comments and contributions. Additionally, we propose allocating future format developments to a dedicated working group under the umbrella of a (to be formed) trapped-charge dating association.

## 8 Conclusions

Our contribution suggested an exchange and long-term data-preservation data format tailored to the specific requirements of the luminescence (dating) community (*XLUM*). The format is *XML*-based and intended to store primary luminescence data and metadata self-consistent. The format implements (but does not enforce) the FAIR guidelines, facilitating a focus on accessibility and findability.

1. On the data storage level, *XLUM* does not constrain the amount of data stored for each measurement by an arbitrary format limitation, i.e. the number of components monitored is not limited by the file format. Furthermore, with this approach, the raw data are self-consistent and inherently contains all relevant information returned by a technical component.

2. On the data analysis level, the format design allows better data quality, as the data wanted for the analysis can be combined with additional information from other technical sensor data. These later data, e.g., stemming from a feedback system monitoring a particular instrument setting, might not be needed to answer the research question; however, it





allows data validation and increases confidence in the result. For instance, failure of technical components may have
invalidated the measurements and have created artefacts. Such records can be excluded in the post-processing.

3. On the data exchange level, data can now be easily exchanged and combined, even though the file format version might
   be modified in the future, which might increase the overall transparency and value of measurement data.

4. On the preservation level, *XLUM* is suitable for long-term preservation of measurement data, because it uses widely used
   and supported standard techniques (*XML*), and it is human-readable.

With *XLUM*, researchers can share their data using a unified file format or make it accessible via publisher websites to
comply with various stakeholder guidelines, such as funding bodies. In the long-term, this approach can pave the way toward
a new type of luminescence data repositories, providing access to primary data on a single component level.

*Author contributions.* **SK** wrote the first draft of the manuscript, contributed to the original format specification and developed and tested its
implementation for R via `'xlum'`. **SG** led the original format design process and wrote down the first format specifications for the *XSYG*-
format. He later developed the Python application 'xlum-python' to provide support for *XLUM*. **MH** contributed to the format specification
and implemented its support as output format for the *lexsyg* TL/OSL readers. **OS** enhanced and refined the original format description.
**KD** raised funds for the development of the predecessor *XSYG* 2013. **GA** Added critical input to initial versions of this manuscript and its
aims. **CB** validated and contributed the `'xlum'` package. **HM** and **GD** contributed to the new format specification in the framework of the
Marie-Curie Action CREDit. All authors discussed and finalised the manuscript.

*Competing interests.* Freiberg Instruments GmbH is a manufacturer of commercial luminescence readers. SK and SG were employed at
the Freiberg Instruments GmbH when the XSYG-file format, the predecessor of XLUM, was specified in early 2013. The initiative for
this manuscript was undertaken independently by SK in the framework of the Marie-Curie Action CREDit to support the distribution of
chronological reference data.

*Acknowledgements.* The *XLUM* file format, has a predecessor, XSYG, originally designed in course of a two months software project in early
2013. This work was supported by the European Union's Horizon 2020 research and innovation programme under the Marie Skłodowska-
Curie grant agreement No 844457 (CREDit).



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
