# Peer review of "XLUM: an open data format for exchange and long-term data preservation of luminescence data"

_Geochronology, 2022_

## Referee Comment (RC3)

XLUM: an open data format for exchange and long-term data preservation of luminescence data

General:

This manuscript is an original, timely and useful contribution to the field of luminescence dating. The paper is well-structured, clearly written and easy to follow even for people who are not familiar with computer scripts. Figures are informative and up to journal standard.

As the authors acknowledge, the success of XLUM will depend on adoption by the community but also on whether journal editors will make it compulsory to include all luminescence data during manuscript submission. The future will tell us. I think there will be broad support in the community and among the manufacturers of luminescence equipment for this initative.

One concern that I have is that in the text (lines 322-324) it is stated that conversion from binx to XLUM is not always lossless. How bad is the loss? 1%? 15% What kind of data is lost? How do the authors envisage the support for the conversion of say binx files to XLUM format in R?

Minor editorial comments:

Line 26: remove 'is' before 'often'

Line 33: insert space before '(Noy…'

Line 42: would suggest to add latest review paper on luminescence dating, Murray et al. Optically stimulated luminescence dating using quartz. Nat Rev Methods Primers 1, 72 (2021). https://doi.org/10.1038/s43586-021-00068-5

Line 53: move 'Fig. 1' to end of sentence

Line 56: subsequence → subsequent

Line 78: emphasis → emphasizes

Line 180: Listing 1 (singular)

Line 340: Listing 4 (singular)

Figure 6: Suggest to remove 'Marie Sklodowska-Curie; Max Karl Ernst Ludwig Planck' from above the figure? It is not clear to me what it does there. Labels to Y-axes?

Line 409: contain instead of contains?

Supporting file: XLUM-file Format Documentation (version 1.0 [2022-09-21])

I have also downloaded and read this document. The whole document requires careful reading by the authors and a check of the English.

My main comment is that Appendix 6 has only the Risø bin/binx to XLUM metadata argument matching. I would expect that also the other dataformats supplied by other manufacturers (see Table 1 in the main paper) are part of such an appendix. Is this not possible? I find it strange that only Risø binx files are discussed.

Page 3, section 3.1, last sentence: I think it is the other way around. Case 1 has three curves and case 2 has 1 curve?

Section 3.1.1: the sentence ' Please note that the XSD document…. (which can be encoded)' does not make sense. Rephrase.

Page 5: In the examples and the Fig. 1 (remove 'the') and start new sentence with 'However'

3.1.3. meaningfully → meaningful; duaration → duration

Table 3: according the… according to?

Page 7: physical information is stored in the…

Table 4: any kind for heating (of heating?); how many kinds of heating are there? Please specify

Isothermal, Infrared (without capital as for the other info)

TM-OSL: … optically stimulated luminescence (can be more specific)

Blue, green, violet,…: add stimulated after colour (e.g. BSL = **b**lue-light **s**timulated **l**uminescence)

Page 7, bottom: change to full sentence 'Valid entries for sampleConditions are given in Table 5.'

Table 5: the information in this table does not describe the sample condition properly. These sample conditions come from the multiple-aliquot (TL) techniques: additive dose procedure; total bleach – additive does procedure; regeneration procedure…

e.g. naturally depleted luminescence to describe "Nat. (Bleach)" is, in my view, possibly incorrect. These aliquots were not necessarily bleached in natural sunlight; a solar simulate lamp was very often used to determine residual dose in MAAD-TL dating of finegrains. See e.g. Wintle A.G., Luminescence dating: laboratory procedures and protocols, Radiation Measurements, Volume 27, Issues 5–6, 1997, Pages 769-817 and references therein.

Tables: make consistent throughout all tables whether information starts with capital or not.

Table 7: doi link overprints information

Table 8: archived

Section 3.6: check second sentence (These instead of There, to ease the sequential of data storage?)

Table 10: manufacturers; unique identifying the parent node?

Section 4: Rephrase third sentence (Either way…)

Section 4.1: Check first paragraph for English (stimulation, data processing…; …. Undertaken processing on the hardware level better?)

Section 5: Practical instead of Pratical

Section 6.1 add comma: block,  others…

p. 14: CURVENO instead of CRUVENO

Remarks: in the remarks column it is written 'considered non-relevant' or 'usage unknown'. However, there must be some relevance or use to these arguments, otherwise it would not have been created. Please improve this and if necessary contact manufacturer.

---

## Author Comment (AC3)

**Response to reviewer #3**

On behalf of my co-authors, I am very grateful for the positive and supportive feedback. Below I will address all comments carefully. Although we are not allowed to upload a new manuscript version at this stage, we considered all comments in our manuscript. Changes referring to the supplementing document are already availabe, because this website is rendered independently of Copernicus.

**Main text**

> One concern that I have is that in the text (lines 322-324) it is stated that conversion from binx to XLUM is not always lossless. How bad is the loss? 1%? 15% What kind of data is lost? How do the authors envisage the support for the conversion of say binx files to XLUM format in R?

We agree. This comment was too thoughtlessly dropped demands elaboration. It refers only to the R package `'xlum'` and not the XLUM format. It boils down to how many resources one should dedicate to subsequent developments *before* the format is finalised. We first wanted to discuss and mature the XLUM format before committing too many resources to the `'xlum'` package development, which would require subsequent work-intensive changes.

Enabling a better conversion is a diligent but routine piece of work. We will iteratively improve the conversion features as soon as we have an initial version accepted (this manuscript, implementation Freiberg Instruments software). We will slightly modify the text, the clarify our approach.

Therefore, the loss cannot and should not be quantified in terms of percentage. It is our aim to make the conversion lossless; which the format allows.

> Line 26: remove 'is' before 'often'

Done.

> Line 33: insert space before '(Noy...'

Done.

> Line 42: would suggest to add latest review paper on luminescence dating, Murray et al. Optically stimulated > luminescence dating using quartz. Nat Rev Methods Primers 1, 72 (2021). https://doi.org/10.1038/s43586-021-00068-5

Agreed.

> Line 53: move 'Fig. 1' to end of sentence

Done

> Line 56: subsequence → subsequent

Done.

> Line 78: emphasis → emphasizes

Done (emphasises)

> Line 180: Listing 1 (singular)

Done.

> Line 340: Listing 4 (singular)

Done.

> Figure 6: Suggest to remove 'Marie Sklodowska-Curie; Max Karl Ernst Ludwig Planck' from above the figure? It is not clear to me what it does there. Labels to Y-axes?

The names are the author names in the example dataset. But we agree that this will unnecessarily lead to confusion. We removed the names in the plot and added y-axis labels ("Component signal [a.u.]").

> Line 409: contain instead of contains?

Agreed. Perhaps it will be changed back later by the editor/copy editor because the non-plural form seems to dominate in modern English.

**Supporting document**

> I have also downloaded and read this document. The whole document requires careful reading by the authors and a check of the English.

We apologise for the typos, we should have been more carefully reviewed this document.

> My main comment is that Appendix 6 has only the Risø bin/binx to XLUM metadata argument matching. I would expect that also the other dataformats supplied by other manufacturers (see Table 1 in the main paper) are part of such an appendix. Is this not possible? I find it strange that only Risø binx files are discussed.

We consider the appendix a guidance, it is not part of the core format description. However, in order to address this comment, we have added similar matching instructions for the `XSYG` and the `PSL` (portable luminescence reader) format.

> Page 3, section 3.1, last sentence: I think it is the other way around. Case 1 has three curves and case 2 has 1 curve?

Thank you for spotting this. We swap the order of the cases.

> Section 3.1.1: the sentence ' Please note that the XSD document.... (which can be encoded)' does not make sense. Rephrase.

Fixed.

> Page 5: In the examples and the Fig. 1 (remove 'the') and start new sentence with 'However' 3.1.3. meaningfully → meaningful; duaration → duration

Done.

> Table 3: according the... according to?

Done.

> Page 7: physical information is stored in the...

Done.

> Table 4: any kind for heating (of heating?); how many kinds of heating are there? Please specify Isothermal,

We removed 'any kind'.

> Infrared (without capital as for the other info)

Done.

> TM-OSL: ... optically stimulated luminescence (can be more specific)

Done.

> Blue, green, violet,...: add stimulated after colour (e.g. BSL = blue-light stimulated luminescence)

Done.

Page 7, bottom: change to full sentence 'Valid entries for sampleConditions are given in Table 5.'

Done.

Table 5: the information in this table does not describe the sample condition properly. These sample conditions come from the multiple-aliquot (TL) techniques: additive dose procedure; total bleach – additive does procedure; regeneration procedure... e.g. naturally depleted luminescence to describe "Nat. (Bleach)" is, in my view, possibly incorrect. These aliquots were not necessarily bleached in natural sunlight; a solar simulate lamp was very often used to determine residual dose in MAAD-TL dating of finegrains. See e.g. Wintle A.G., Luminescence dating: laboratory procedures and protocols, Radiation Measurements, Volume 27, Issues 5–6, 1997, Pages 769-817 and references therein.

Thank you for this comment, we corrected it in the document.

Tables: make consistent throughout all tables whether information starts with capital or not.

Done.

Table 7: doi link overprints information

The PDF is auto-generated, the overprinting does not happen in the HTML, which is the main document, the PDF is only for convenience. Tweaking this minor issue would be rather complex or prompt a wrong examples (e.g., shorten but invalid DOI)

Table 8: archived

Done.

Section 3.6: check second sentence (These instead of There, to ease the sequential of data storage?)

Done.

Table 10: manufacturers; unique identifying the parent node?

Done.

Section 4: Rephrase third sentence (Either way...)

Done.

Section 4.1: Check first paragraph for English (stimulation, data processing...; .... Undertaken processing on the hardware level better?)

Done.

Section 5: Practical instead of Pratical

Thanks, done.

Section 6.1 add comma: block, others...

Done

p. 14: CURVENO instead of CRUVENO

Done.

Remarks: in the remarks column it is written 'considered non-relevant' or 'usage unknown'. However, there must be some relevance or use to these arguments, otherwise it would not have been created. Please improve this and if necessary contact manufacturer.

Indeed, manufacturer data format usually includes additional bits necessary for other reasons but are irrelevant for the long-term storage of luminescence data. Still, they can be stored in the XLUM format, but the

arguments are non-compulsory. This may, of course, change in the future. Using those arguments would not render the XLUM file format incompatible. It only indicates that parsers do not need to process them.

On behalf of all co-authors,

Sebastian Kreutzer, Heidelberg, 2023-03-2

---

## Author Response (AR1)

**Response to Pieter Vermeesch (Associate Editor)**

> Both the reviewers and myself agree with the need to record raw data in a human-readable ASCII format. You have chosen an XML format, but you could have also chosen JSON. In fact, line 363 of the manuscript points out that "JSON has gained popularity over XML in recent years". So I am curious why you used XML. The example of Section 5 would look cleaner in JSON than XML.

Thank you for this particular question, which we thought would come earlier. In essence, we had three reasons:

1. The predecessor of XLUM is, to some extent, the XSYG format developed years ago in the framework of another software project at Freiberg Instruments. While this format did not gain much attention over the last ten years, there was still some room to test the format and spot flaws. This experience ultimately led to the proposal of the XLUM format.

2. XML is highly standardised and used countless times, particularly in data archives. It is sometimes considered a little bit verbose. Still, the available parsers for XML and JSON are not much different in speed, making it more of a personal preference. But we agree that it will be desirable to have a JSON conversion in the future.

3. We aim to get **the** first format definition out and accepted by the community. If we reach that aim, it will be straightforward to convert data into other formats if needed.

> For example, OSL timestamps and signals are stored in vectors of equal length. The XML code places these two vectors at different hierarchical levels of the 'curve' tags. Using an abridged version of the example: 293 303 313 323 333 343 353 363 373 383 JSON would allow you to store the same values in a two-dimensional array, which better represents the information content of the data. For example:

```
curve: {
duration: 10,
signal: [[1,2,3,4,5,6,7,8,9,10],[293,303,313,323,333,343,353,363,373,383]]
}
```

This is correct, and perhaps more is owed to a personal preference. Here we preferred to separate data from metadata. The data measured by the component are always and exclusively stored in the `<curve/>` node. Everything else we tagged as metadata (including the timestamps).

> Since it is quite easy to convert JSON to XML and vice versa, I would suggest that you consider providing both options. Or instead of specifying a specific format, perhaps you could simply propose a schema that can be applied to any database format?

As you wrote, it is easy to convert between both formats. The key to our proposal is the data structure emphasises component-based data storage and the proposed design. If we were to propose this as XML and JSON simultaneously, this may lead to confusion and require some extra effort we think is not justified at the current stage. If JSON makes it easier (in particular regarding databases): certainly we will write a converter and then propose XLUM as XML and JSON format. But we are not yet there, so first things first.

I hope our reasoning makes somewhat sense. It is not about avoiding the effort, but about communicating a standardised format with the solid basis but without further complicating the approach in the first place.

> Line 7: too many commas

Done.

> Line 21: typo -> "Rechereche"

Done.

> On line 67, there is a sudden change of pace. Whereas the previous paragraphs required no specialised knowledge of OSL, point 2 of this enumerated list suddenly moves on to advanced R packages without properly introducing them. I also wonder if the list really needs 5 items, since items 2 and 3 seem to make broadly the same point.

Thank you for pointing this out. We merged items 2 and 3 and added one more line for the cited R packages. This does not really detail the R packages but presents their idea in a nutshell.

> Line 104: Please avoid double brackets between references.

Now removed. The double brackets were in place to avoid the impression that *Zenodo* was developed by us.

> Line 105: Replace "At last some format conventions, hereafter we will" with "In the remainder of this paper, we will" or something along those lines.

Done.

> Figure 2: What is the purpose of the equation? Either explain it or remove it. Caption of Figure 2: "the minimum and maximum values"... of what?

The equation details the calculation andv "balances" the figure (three panels in each column). It is now mentioned in the figure caption, where we also added "temperature" for the "the minimum and maximum values".

> Section 3 contains a long list of requirements, several of which are redundant. I suggest replacing all instances of "should" and "shall". For example, replace "The format should preserve..." with "The format preserves...".

We removed "should" and "shall" (both expressions used technical requirement documents) and merged the last two points.

> Line 162: remove "nearly"

Done.

Line 165: rephrase "2003, the year of the article by Bortolot and Bluszcz (2003);"

Done.

Line 219 and Equation 1: what is the relationship between 'v' and 'A'?

$v$ spans the array $A$ (now written in the text)

Line 222: instead of using two levels of subscript (e.g., `$A_{(2,1)}_1$`), I suggest using 3-dimensional arrays (e.g., $A_{(2,1,1)}$).

Agreed; I cannot remember anymore why we had chosen this, but your suggestions reads less complicated.

Figure 6: What is the difference between the top and bottom panel? Both have the same horizontal axis and are labelled "PMT-luminescence in [cts] 'measured'".

We added a y-axis and added more information to the figure caption.

Line 380: "The notion of open data is access and insight" is not a proper sentence.

Corrected.

In general, Section 7 is too wordy and vague. Please sharpen and shorten.

We agree, we shortened the text and sharpen the wording.

*Final note: we had responded to the reviewers' comments in the public discussion, all comments have been addressed.*

On behalf of all co-authors,

Sebastian Kreutzer, Heidelberg, 2023-03-07